# CRISPR/Cas9 Technology and Its Utility for Crop Improvement

**DOI:** 10.3390/ijms231810442

**Published:** 2022-09-09

**Authors:** Hua Liu, Wendan Chen, Yushu Li, Lei Sun, Yuhong Chai, Haixia Chen, Haochen Nie, Conglin Huang

**Affiliations:** 1Beijing Academy of Agriculture and Forestry Sciences, Beijing 100097, China; 2Beijing Key Laboratory of Forest Food Processing and Safety, Department of Food Science and Engineering, College of Biological Sciences and Biotechnology, Beijing Forestry University, Beijing 100083, China; 3Beijing Vocational College of Agriculture, Beijing 100097, China

**Keywords:** CRISPR/Cas9, gene editing, crop improvement, plant breeding, gene function

## Abstract

The rapid growth of the global population has resulted in a considerable increase in the demand for food crops. However, traditional crop breeding methods will not be able to satisfy the worldwide demand for food in the future. New gene-editing technologies, the most widely used of which is CRISPR/Cas9, may enable the rapid improvement of crop traits. Specifically, CRISPR/Cas9 genome-editing technology involves the use of a guide RNA and a Cas9 protein that can cleave the genome at specific loci. Due to its simplicity and efficiency, the CRISPR/Cas9 system has rapidly become the most widely used tool for editing animal and plant genomes. It is ideal for modifying the traits of many plants, including food crops, and for creating new germplasm materials. In this review, the development of the CRISPR/Cas9 system, the underlying mechanism, and examples of its use for editing genes in important crops are discussed. Furthermore, certain limitations of the CRISPR/Cas9 system and potential solutions are described. This article will provide researchers with important information regarding the use of CRISPR/Cas9 gene-editing technology for crop improvement, plant breeding, and gene functional analyses.

## 1. Introduction

The rapidly growing global population has led to a substantial increase in the demand for food crops. However, traditional crop breeding methods require considerable labor and resources and are time consuming. Thus, the available crop varieties and current agronomic practices will not be able to meet the future needs for the global demand for food. Moreover, emerging crises leading to decreased crop yields and quality are exacerbated by subtle climate changes and the loss of natural genetic resources. In this context, the development and application of gene-editing methods, which can rapidly modify crop traits via precise genetic modifications, is a promising approach for accelerating the improvement of germplasm resources and the generation of new varieties to meet the challenges of feeding the world [1]. The recent emergence of gene-editing technologies has provided researchers with powerful tools for decoding gene functions and enhancing plant traits through diverse biological systems involving zinc finger nucleases (ZFNs), transcription activator-like effector nucleases (TALEN), and clustered regularly interspaced short palindromic repeat (CRISPR)/CRISPR-associated protein (CRISPR/Cas) systems. Gene-editing technologies enable the modification of a specific genomic site (e.g., gene knockout/single-base editing and guided editing) using artificially designed nucleases [2].

The CRISPR/Cas system has been widely used to edit plant genomes and create mutants because of its simplicity and convenience. It is increasingly being used for modifying the traits of many plants, including important crops, and for developing new germplasm resources [3]. The CRISPR/Cas9 system commonly used for genome editing involves the cleavage of DNA by the RNA-guided DNA endonuclease Cas9. The optimization of this system facilitates the efficient and accurate modification of target genes, thereby accelerating plant breeding. Consequently, CRISPR/Cas9 has gradually become the most widely used and advanced gene-editing system [4]. This review introduces the CRISPR/Cas9 genome-editing system by describing the progress in the associated research, the underlying mechanism, the related technology, and its utility for crop breeding. Furthermore, the limitations of the CRISPR/Cas9 system and how they may be overcome are discussed. This review provides researchers with critical information relevant for applying CRISPR/Cas9 gene-editing technology to enhance crops and breed novel cultivars.

## 2. Development of CRISPR/Cas9 and Clarification of the Underlying Mechanism

The CRISPR/Cas system is an adaptive immune defense mechanism that evolved in bacteria and archaea. In 1987, the Nakata research group at Osaka University (Japan) detected five 29-nucleotide repetitive palindromes in the 3′ flanking sequence of the *iap* gene in *Escherichia coli* [5]. These repeats (i.e., CRISPR) have since been detected in a variety of prokaryotes. The genes flanking CRISPR loci (*Cas1* to *Cas4*) are reportedly associated with the bacterial immune system [6,7,8]. In 2007, Barrangou et al. conducted viral challenge experiments, which revealed CRISPR/Cas in bacteria-mediated viral resistance [9]. Subsequent research has confirmed the importance of the CRISPR/Cas system for the bacterial immune defense mechanism [10]. The CRISPR/Cas locus consists of the following three elements: a *trans*-activating RNA (tracrRNA) gene at the 5′ end, a Cas protein-encoding gene in the middle, and a CRISPR locus at the 3′ end, which is mainly composed of a lead sequence followed by a 23–50 bp repeat sequence and a spacer sequence. The interval sequence consists of 17–84 bp, with an average length of approximately 36 bp [11].

The CRISPR/Cas systems that have been identified can be divided into two categories on the basis of the number of Cas proteins (Class I and Class II) and subdivided into six types (Type I to Type VI) according to the structure and function of the Cas proteins. Class I includes Type I, Type III, and Type IV, whereas Class II includes Type II and Type V [12]. In contrast to Class I systems, those in Class II require only one Cas protein. Hence, the current commonly used CRISPR/Cas gene-editing systems belong to Class II, including those involving Cas9, Cpf1 (Cas12a) without a tracrRNA, and Cas13 with RNA-cutting activity [13]. Additionally, Cas9 contains the HNH domain and the RucV-like domain. The HNH domain cuts complementary strands of CRISPR RNA (crRNA), whereas the RucV-like domain cuts non-complementary strands, resulting in DNA double-strand breaks (DSBs) [14]. Unlike the Cas9 protein, Cpf1, which is the DNA endonuclease in a novel CRISPR-based system, contains a RuvC domain and lacks the HNH domain [15]. Moreover, the editing of the genome by Cpf1, which uses a guide RNA (gRNA) that is significantly different from that used by Cas9, may be regulated at the post-translational level.

The fact that it specifically recognizes and cuts DNA to produce DSBs makes the CRISPR/Cas9 system appropriate for editing target genes. In 2012, the Doudna and Charpentier research group at the University of California (Berkeley, CA, USA) demonstrated the CRISPR/Cas9-specific cleavage of target DNA in vitro. In addition, crRNA–tracrRNA was converted into a single-strand guide RNA (sgRNA) [4]. In 2013, Feng Zhang of the Massachusetts Institute of Technology and George Church’s research group at Harvard University were the first to describe the use of CRISPR/Cas9 for editing genes in mammalian cell lines [16,17]. Laboratories worldwide have subsequently exploited this new gene-editing tool (Figure 1).

After more than 20 years of research, the CRISPR/Cas9 mechanism of action is relatively clear. In the CRISPR/Cas9 system, tracrRNA and crRNA combine to form a complex that recruits and guides Cas9 to cut the DNA sequence at a specific genomic location. Researchers went a step further and converted the tracrRNA and crRNA complex into sgRNA, which consists of only one RNA strand. Guided by sgRNA, the Cas protein recognizes a conserved sequence [18,19]. Specifically, it recognizes and binds to the protospacer adjacent motif (PAM) and unwinds the double-stranded DNA. The crRNA is a complementary sequence and pairs with the target sequence upstream of the PAM. The crRNA and tracrRNA, which are mature products derived from the CRISPR locus, form a sgRNA through base complementation and pairing. The sgRNA is paired with the sequence upstream of the PAM, resulting in a DSB [14]. The DSBs are repaired by non-homologous end joining (NHEJ) or homology-directed repair (HDR). The NHEJ repair method is prone to errors, with small fragments deleted or inserted at the break site, resulting in gene mutations. In the presence of donor DNA, the break site is repaired via HDR, which precisely inserts or replaces bases (Figure 2) [20,21,22].

The CRISPR/Cas9 system and its related gene-editing technologies have developed rapidly and have been widely used for the functional characterization of crop genes and for precise molecular design-based breeding. It will also be important for future crop genetic engineering and breeding [23].

## 3. CRISPR/Cas9-Based Gene-Editing Tools

In recent years, CRISPR/Cas9 has been among the most flexible systems used for modifying specific genes in diverse genomes. Many CRISPR/Cas9-based tools have been developed, enabling researchers to modify genes in various ways (e.g., gene knockout, gene knock-in, gene regulation, base editing, and prime editing) [3].

### 3.1. Gene Knockout

Gene knockout methods are crucial for analyzing and verifying gene functions and the associated changes to biological traits. For example, CRISPR/Cas9 gene knockout vectors are usually constructed and transferred into plants. After screening for one or two generations, target-gene knockout mutants may be obtained [24]. One target gene can be knocked out or multiple target genes can be knocked out simultaneously. Multiple sgRNAs can be incorporated into a binary vector using Golden Gate, Gibson, or isotail polymer techniques [25]. The gene-editing efficiency can be improved by targeting multiple sites in a single gene of interest [26].

### 3.2. Gene Knock-In

Gene knock-in methods are used to insert an exogenous DNA fragment into a specific genomic locus after cleaving the DNA with Cas. Following the introduction of a precisely determined DSB into the genome, depending on the characteristics of the donor vector and the cell cycle phase, two major DNA repair pathways (HDR and NHEJ) are used. According to the cellular repair pathways, CRISPR-based gene knock-in methods mainly involve either homology-independent strategies (NHEJ-based targeted insertion events) or homology-dependent strategies (HDR-based targeted insertion events), further elucidating that the cellular DNA break repair pathways may enhance the utility and efficiency of gene insertion methods [27].

### 3.3. Base Editing

Base editors comprise single-base editors and double-base editors. The initial base editors were cytosine base editors (CBEs) and adenine base editors (ABEs), both of which are single-base editors mediating only one type of base transition (i.e., C-to-T and A-to-G, respectively). Thus, their utility for site saturation mutagenesis is limited [28]. The CBE system is composed of sgRNA, a nickase Cas9 (nCas9) that cleaves a single-stranded sequence, cytosine deaminase, and uracil glycosylase inhibitor (UGI). The mechanism underlying the CBE system is as follows: guided by the sgRNA, nCas9 cleaves the non-target strand, whereas cytosine deaminase modifies a cytosine (C) residue in the target-gene editing window by removing an amino group and converting the residue to uracil (U), the removal of which is prevented by UGI. During DNA replication, U is recognized as thymine (T), which is complementary to adenine (A). In the subsequent round of DNA replication, A and T are paired normally to convert C to T [29]. In contrast, the basic components of the ABE system are sgRNA, nCas9 that cleaves a single-stranded sequence, and adenine deaminase fused to nCas9. The mechanism underlying the ABE system is as follows: under the guidance of the sgRNA, nCas9 cuts the non-target strand, after which the A in the target-gene editing window is converted to inosine (I) following the removal of the amino group in a reaction catalyzed by adenine deaminase. During DNA replication, the modified residue is recognized as guanine (G), which is complementary to C. In the next round of DNA replication, G pairs with C, thereby completing the A-to-G transition [30]. Double-base editors, such as saturated targeted endogenous mutagenesis editors, can convert both C and A to G at target sites under the guidance of the sgRNA, thereby significantly increasing the saturation and diversity of base mutations [31,32]. A new glycosylase base editor (CGBE) system was recently reported, in which UGI is replaced by U-DNA glycosylase (UNG); this system efficiently induces the targeted C-to-G base transversion as well as C-to-T and C-to-A conversions [33,34,35]. In another study, which revealed that an interbacterial toxin (i.e., DddA) catalyzes the deamination of cytidines, RNA-free DddA-derived CBEs (DdCBEs) were developed to facilitate the targeted C/G-to-T/A conversions within mitochondrial DNA (mtDNA) [36]. The DdCBEs have been used to edit bases in plant mtDNA [37].

The utility of CBEs and ABEs helped make CRISPR/Cas-based genome-editing tools more widely available [38]. Methods involving these base editors can modify bases in genomes without inducing DSBs and do not involve NHEJ/HDR. Human APOBEC3a-based editors can efficiently convert C to T in wheat, rice, and potato [38]. Furthermore, a seventh-generation ABE (7.10) was observed to effectively induce the A-to-G conversion in rice and wheat, with an efficiency of up to 60% [30,39].

### 3.4. Prime Editing

In 2019, David Liu developed prime editing (PE) as a more accurate gene-editing method that can edit genes without generating DSBs or introducing donor DNA. This method, which enables 12 types of base substitutions, was developed by modifying the CRISPR/Cas9 system. First, nCas9 (H840A) was combined with *Moloney murine leukemia virus* (M-MLV) reverse transcriptase to form a new fusion protein. Second, an RNA sequence containing a primer-binding site and a reverse transcription template was added to the 3′ end of the sgRNA to produce the prime editing-extended guide RNA (pegRNA). Finally, under the guidance of the pegRNA, the nCas9 (H840A) and M-MLV reverse transcriptase complex is brought to the target site, where a single-stranded DNA sequence containing the site-directed mutation is generated by the reverse transcriptase using the reverse transcription template of the pegRNA. Various precise gene mutations were obtained through the DNA repair pathway [40]. The basic components are pegRNA, nSpCas9 (H840A), and M-MLV reverse transcriptase. The pegRNA is a modified gRNA, with a primer-binding site and a reverse transcription template at the 3′ end; this template provides additional information required for editing. The nSpCas9 enzyme is derived from Cas9 and can only cleave single-stranded DNA. During PE, the gRNA is paired with the target gene, which guides nSpCas9 to cut the target strand, leading to a single-strand break. The BS is connected to the 3′ end of the notch, bringing the reverse transcription template to the notch. Next, the M-MLV reverse transcriptase synthesizes single-stranded DNA sequences from the 3′ end of the fracture using the reverse transcription template. This triggers the automatic repair mechanism, which uses the newly synthesized DNA sequence as a template to generate another DNA strand to introduce a base substitution at any position of the DNA double strand [41]. The ability to modify all bases (including conversions and transpositions) without the need for DNA templates and DSBs makes PE a very safe and potentially useful gene-editing method. In 2020, Gao Caixia’s research team developed the Plant Prime Editor, which can mediate diverse single-base substitutions in rice and wheat [42].

### 3.5. Gene Regulation

Gene regulation includes transcriptional regulation and post-transcriptional regulation, with transcriptional regulation including DNA regulation at the genetic level and chromatin regulation at the epigenetic level [43]. Death Cas9 (dCas), which was obtained by modifying the Cas protein, lacks nuclease activity, but can recognize specific DNA sequences [44]. The binding of dCas to double-stranded target genes is similar to the binding of transcription factors to target gene promoters; however, dCas cannot function alone, but it can cause steric hindrance. Researchers have combined dCas and transcriptional activators/suppressors to modulate gene transcription [45]. When dCas binds to the promoter or transcription start site of the target gene, it can prevent the initiation of transcription. When dCas binds to the target gene open reading frame, it prevents the binding of RNA polymerases and transcription factors, thereby inhibiting transcription [46].

Targeting the promoter region is an effective way to regulate gene expression [47]. For example, in an earlier study, the transcriptional activator VP64 and dCas were fused and the resulting complex was bound to the CpG methylation site C in the *AtFIS2* promoter region in *Arabidopsis thaliana*, which eliminated the inhibitory effects of CpG methylation on transcription [48]. In addition, CRISPR/Cas9 functions via epigenetic regulation involving chromatin, gene modifications following the binding of dCas to DNA methylase and acetylase, or by altering the chromatin structure and modulating the interaction between the enhancer and promoter to regulate gene expression [49].

## 4. Application of the CRISPR/Cas9 System for Crop Breeding

The basic goal of research related to plant genetics and breeding is to elucidate the association between the genotype and phenotype. Traditional cross breeding mainly relies on phenotypic observations and the experience of the breeder to select enhanced varieties. Important agronomic traits are generally controlled by multiple quantitative loci. Moreover, there is some correlation between different agronomic traits, and modular gene regulation is common. This complexity is a major challenge to traditional cross breeding [50]. The development of high-throughput sequencing technology has resulted in the increasing availability of sequenced crop genomes, which has greatly promoted the study of gene functions and the mining of genes regulating important traits, including yield, quality, stress tolerance, and disease resistance. Key regulatory genes and the associated networks controlling complex crop traits have been identified through gene function-related research. Furthermore, some of these genes have been accurately edited to enhance germplasm resources and gradually establish accurate molecular breeding systems. In this context, CRISPR/Cas9 can be used to improve a variety of crop traits, such as yield, quality, stress tolerance, disease resistance, and herbicide resistance, to create a substantial abundance of new germplasm.

### 4.1. Improvement of Crop Disease Resistance

In recent years, there has been considerable research on the application of the CRISPR/Cas9 system to increase crop resistance to fungi, bacteria, and viruses. Typically, the following strategies are used to edit the genome and alter specific plant defense mechanisms to improve plant disease resistance.

#### 4.1.1. Modification of R Genes

The main methods for modifying R genes include editing the pathogen recognition sites encoded by the known R genes to improve recognition, replacing the recognition-related region of R genes to enable the recognition of nonspecific pathogen effectors, and increasing R gene expression levels.

#### 4.1.2. Modification of S Genes

The common methods used to modify S genes involve mutating the effector recognition site encoded by the S gene, removing or inactivating negative immune regulatory factors, and inhibiting S gene expression.

#### 4.1.3. Targeted Degradation of Viral Genomes

The CRISPR/Cas system may be incorporated into a crop, wherein it specifically cuts viral DNA or RNA.

The CRISPR/Cas-based editing tools have been used to increase plant resistance to fungal diseases. A loss-of-function mutation to *MILDEW LOCUS O* (*MLO*) leads to increased powdery mildew resistance; this gene was first identified in barley and was subsequently revealed as a typical S gene in monocotyledons and dicotyledons [51]. Studies on wheat and tomato have shown that mutations in the *MLO* gene can lead to disease resistance. For example, in an earlier investigation of three *MLO* homologous alleles in wheat, the CRISPR/Cas9-induced mutation to *TaMLO-A1* enhanced wheat resistance to powdery mildew [52]. Sixteen *SlMLO* alleles were identified in tomato, of which *SlMLO1* is the most important for disease resistance. Two similar sites in *SlMLO1* were cut using the CRISPR/Cas9 system, resulting in the deletion of a 48-bp DNA fragment from *SlMLO1*. A new non-transgenic tomato variety (‘Tomelo’) highly resistant to powdery mildew was developed by selfing. Off-target analyses indicated that the genomic regions beyond the *SlMLO1* locus were unaffected [53]. Using SlU6-2P4 as the promoter to drive the sgRNA, we constructed a CRISPR/Cas9 genome-editing vector targeting the powdery mildew resistance-related genes *MLO1* and *EDR1* in tomato [54]. In other studies, rice blast-resistant mutants were generated by knocking out *OsERF922* and *OsSEC3A* using the CRISPR/Cas9 system [55,56]. Four gRNA-targeted knockdowns of the transcription factor gene *VvWRKY52* increased the resistance of grape plants to *Botrytis cinerea*, but there were no other significant phenotypic differences between the mutant plants and the wild-type control [57]. The aforementioned examples confirm that CRISPR/Cas editing technology can effectively improve crop resistance to fungal diseases.

In addition to fungal diseases, bacterial diseases also seriously affect crop yield and quality. In rice, *Xanthomonas oryzae* pv. *oryzae* (Xoo) infects plants by inducing the expression of a sucrose transporter gene. The CRISPR/Cas9 genome-editing system has been used to simultaneously edit the promoter regions of *Sweet11*, *Sweet13*, and *Sweet14* to obtain rice lines exhibiting broad-spectrum resistance to multiple Xoo physiological races [58]. In grapefruit, a mutant resistant to the citrus ulcerative pathogen *Xanthomonas citri* subsp. *citri* (Xcc) was obtained by editing the PthA4 effector-binding element in the *CsLOB1* promoter [59]. Similarly, Jia et al. used CRISPR/Cas9 technology to remove the effector-binding region of the *CsLOB1* promoter and obtain mutants with significantly enhanced resistance to Xcc [60]. Malnoy et al. used CRISPR/Cas9 ribonucleoproteins (RNPs) to edit *DIPM-1*, *DIPM-2*, and *DIPM-4* in apple protoplasts, which led to increased fire blight resistance [61]. These studies indicate that CRISPR/Cas-based editing technology can be applied to improve crop resistance to bacterial diseases.

Viral diseases can also significantly decrease crop yield and quality. Accordingly, CRISPR/Cas9 has been used to increase crop resistance to viral diseases. Viral genomes vary considerably and may comprise double-stranded DNA (dsDNA), single-stranded DNA (ssDNA), double-stranded RNA (dsRNA), and single-stranded RNA (ssRNA). By targeting DNA or RNA, CRISPR/Cas systems can directly degrade viral genomes. Specifically, CRISPR/Cas gene-editing techniques involving Cas9 or Cas13a have been applied to improve crop resistance to DNA or RNA viruses [51,62]. We previously designed 11 sgRNAs that target the sequence encoding the replication initiation protein (Rep) motif of *Soybean yellow dwarf virus* (BeYDV) as well as the Rep-binding site, hairpin structure, and non-nucleotide sequence to decrease the viral load in tobacco by 87% [63]. Ali et al. developed sgRNAs targeting conserved stem-loop sequences specific to the coding and non-coding sequences of *Tomato yellow leaf curly virus* (TYLCV) to significantly restrict the replication and accumulation of the virus [64]. Two variants of Cas9, *Francisella novicida* Cas9 (FnCas9) and Cas13a, reportedly can directly target and degrade RNA. Moreover, RNA-targeting sgRNA and FnCas9 vectors for *Cucumber mosaic virus* (TMV) were constructed and expressed in tobacco and *A. thaliana*. The accumulation of CMV and TMV in the resulting transgenic lines decreased by 40% to 80% compared with the control. Furthermore, the resistance achieved by the sgRNA–FnCas9 system was stably inherited [65].

Another option involves modifying the antiviral genes of crops. Exogenous Cas-encoding genes and sgRNA are continuously expressed in the plant after the viral genome is edited, ultimately leading to increased protection against phytopathogenic viruses. Therefore, it will be subject to strict supervision under genetic modification safety policies. Non-transgenic varieties resistant to viruses can be obtained by editing crop antiviral genes. RNA viruses usually exploit host plant regulators, such as the eukaryotic translation initiation factors eIF4E, eIF(iso)4E, and eIF4G, to complete their life cycle [51]. Using CRISPR/Cas9, researchers targeted two sites in the cucumber susceptibility gene *eIF4E* to generate mutant plants, after which the CRISPR/Cas9 vector in the genome was removed by backcrossing. The resulting plants were resistant to *Potato virus Y* (Potyviridae), *Cucumber pulse yellow mosaic virus* (CVYV), *Zucchini yellow mosaic virus* (ZYMV), and *Papaya dot mosaic virus-W* (PRSV-W). Macovei et al. developed a rice strain resistant to *Rice tungro spherical virus* by mutating the *eIF4G* allele with a CRISPR/Cas9 system [66]. Knocking out *StDND1*, *StCHL1*, and *StDMR6-1* (DMG400000582) using a CRISPR/Cas9 system generated potatoes with increased resistance to late blight [67]. A recent study determined that enhanced *Capsicum annuum* anthracnose resistance may be achieved via the CRISPR/Cas9-mediated alteration of the susceptibility gene *CaERF28* [68]. The examples of genes from various crops that have been modified by the CRISPR/Cas9 system to increase disease resistance were shown in Table 1.

### 4.2. Improvement of Crop Herbicide Tolerance

Weeds greatly limit crop growth and yield. Various effective herbicides have been developed that mitigate the adverse effects of weeds and substantially increase the yield of grains and other crops. Most herbicides target critical plant-specific metabolic enzymes to kill plants. The development of herbicide-tolerant crops able to withstand specific herbicides can increase the utility of herbicides. Compared with traditional breeding methods, the use of CRISPR/Cas technology can accelerate the creation of crops tolerant to multiple herbicides [69].

Acetolactate synthase (ALS) is a crucial enzyme for the biosynthesis of branched amino acids. Multiple herbicides targeting ALS have been developed, including sulfonylurea and imidazolinone herbicides [70]. Due to the conserved nature of *ALS* genes, tolerant mutant lines can be obtained by substituting bases in *ALS*. As the initial base-replacement pathway, HDR requires exogenous templates that are homologous to the target site as well as RNA as the repair template to repair DSBs in different species.

Several herbicide-tolerant materials have been created by using CRISPR/Cas to edit the *ALS* gene in model plants, rice, wheat, and other crops. Studies of naturally occurring point mutations in the *A. thaliana ALS* gene suggest that substituting specific bases in *ALS* may lead to herbicide tolerance [71]. The sgRNA can be inserted into target cells relatively easily to serve as the repair template. Hence, Butt et al. designed a chimeric sgRNA (cgRNA) that functions as both the sgRNA and the repair template [72]. Sixty-seven cgRNA structures varied regarding the resulting editing efficiencies. The targeted editing of *OsALS* using a cgRNA/Cas9 gene-editing platform quickly and efficiently generated rice lines tolerant to bispyribac-sodium. The targeted editing of *OsALS* by CBE or ABE systems can also confer herbicide tolerance to rice [39,73]. Zhou Huanbin’s research team (Kuang et al., 2020) mutated *OsALS1* and *OsACC* by editing single bases, and successfully created the herbicide-tolerant rice variety ‘Nanjing 46’ [74].

The wheat *TaALS* gene can be edited by CBE to produce wheat mutant lines able to grow after being treated with herbicides [75]. Zhang et al. used base-editing technology to develop transgene-free wheat germplasm tolerant to sulfonylurea, imidazoline ketone, and aryloxyfluorophenoxy propionic acid herbicides. In addition, the generation of wheat plants tolerant to nicosulfuron herbicides enabled the selection of wheat capable of tolerating two herbicides [76]. Veillet et al. targeted the potato *StALS* gene using gene-editing techniques, with a success rate of 92% [77].

Mutating the *Brassica napus BnALS* gene by CBE reportedly leads to herbicide tolerance. In watermelon, *ALS* was modified by base editors to obtain herbicide-tolerant transgene-free watermelon [78]. Similarly, herbicide-tolerant soybean germplasm was generated by editing *ALS* [79].

The CRISPR/Cas9 system was used to accurately replace *ZmALS2* in maize to produce chlorsulfuron-tolerant plants. Target-AID, which is a synthetic complex formed via the fusion of dCas and PmCDA1, can mediate the C-to-T substitution at specific genomic targets in yeast and mammalian cells [39]. The incorporation of nCas9 (D10A) can increase the efficiency of the Target-AID system. The codon optimization of Target-AID makes it suitable for editing plant genomes. For example, it has been used to accurately edit the rice *ALS* gene to generate herbicide-tolerant plants [80]. The ABE7.10 system can precisely convert A/T to G/C. Gao Caixia and co-workers used this system to introduce a point mutation at position 2186 of the rice *OsACC* gene to replace a Cys residue with an Arg residue in the corresponding protein, thereby creating herbicide-tolerant rice [28]. A novel artificial rice germplasm resistant to dinitroaniline herbicides was developed by altering the *OsTubA2* sequence [81].

### 4.3. Improvement of Crop Yield

Increasing yield is a major objective of studies aimed at improving crops. Specific grain yield-related traits include grain number and size per panicle, tiller number per panicle, grain weight, and grain size [38,82]. Recent research has revealed genetic modifications that have increased the yield of rice, wheat, and other food crops on the basis of CRISPR/Cas9 technology (Table 2). Miao et al. generated a *pyl1/4/6* triple knockout rice mutant using the CRISPR/Cas9 system. Compared with the wild-type control, the mutant had a higher yield, longer panicles, more primary and secondary branches in the panicles, and fewer tillers per plant [83]. In a different study, the rice grain weight increased significantly following the simultaneous deletion of three grain weight-associated genes (*GW2*, *GW5*, and *TGW6*) [84]. Knocking out *OsAAP3*, which encodes an amino acid transporter associated with the allocation of nutrients in rice, reportedly increases the number of tillers and the grain yield, while maintaining the grain quality [85]. Additionally, knocking out *OsSNB*, which helps regulate plant development (e.g., floral organ formation) [86], using the CRISPR/Cas9 system can increase the grain length and width as well as the 1000-grain weight, implying that in addition to its effects on floral development, OsSNB also controls the rice grain shape [87]. Zhang et al. (2016) targeted *TaGASR7* by transiently expressing CRISPR/Cas9 DNA or RNA in the calli of hexaploid wheat and tetraploid durum wheat, and observed that the 1000-grain weight increased in the T_0_ mutant [88].

Deleting the *TaGW2* gene encoding a RING E3 ligase in wheat increases the grain length and width, thereby increasing the grain yield [89,90]. Among the factors affecting grain yield, modulating cytokinin homeostasis may be an effective strategy for improving the grain yield. More specifically, knocking out *TaCKX2-D1*, which encodes a cytokinin oxidase/dehydrogenase in wheat, increases the grain yield [91]. The number of grains, the grain weight per spike, and the total rice yield increase after the CRISPR/Cas9-mediated knockout of the 3′ terminal of the *OsLOGL5* coding region [92]. Wang et al. identified *GRAIN SIZE ON CHROMOSOME 2* (*GS2*) and designed a novel gene-editing method that can be widely employed to increase the rice grain size and yield. They also suggested that this method for improving crops is applicable for other genes containing miRNA target sites, especially the conserved miR396-*GRF*/*GIF* module that influences plant growth, development, and responses to environmental stimuli [93].

### 4.4. Improvement of Crop Quality

With the general improvement in global living standards, the demand for high-quality crops is increasing. The market value of crops is greatly influenced by crop quality, which is determined by external and internal traits. Physical characteristics, including size, color, and texture, as well as fragrance, are important factors affecting crop quality. The contents of specific nutrients (e.g., proteins, starch, and lipids) and bioactive substances (e.g., carotenoids, lycopene, γ-aminobutyric acid, and flavonoids) influence the internal quality-related crop characteristics [94].

Recent research involving CRISPR/Cas9 gene-editing technology has resulted in considerable improvements in crop quality. Grains with low amylose contents are preferred because of their nutritional and cooking value. They are also widely used in the textile and adhesive industries. The CRISPR/Cas9 system can be used to decrease the amylose content and improve the nutritional value and flavor of rice grains. Ma et al. successfully reduced the amylose content of rice grains from 14.6% to 2.6% by knocking out the *WAXY* (*Wx*) gene, thereby obtaining *waxy* rice mutant plants [25]. Similarly, the maize *Wx1* gene encodes a starch synthase affecting the grain composition [46]. Knocking out *Wx1* via the CRISPR/Cas9 system can increase the amylopectin content of maize grains to almost 100% [47], without inducing other phenotypic changes. Sun et al. (2017) used CRISPR/Cas9 technology to knock out the *SBEIIb* gene; the examination of the *SBEIIb* mutants indicated the resistant starch content increased from less than 1% to 9.8% [95,96,97]. Shiting Fan reported that the targeted deletion of the *WAXY* coding region in spring barley using the CRISPR/Cas9 system can generate lines with decreased amylose contents, which positively affects the edibility and processing quality of barley grains [98]. Flavor is an important quality of rice grains. Commercially valuable cooked rice varieties with superior flavors are readily available. Knocking out the gene encoding betaine aldehyde dehydrogenase (*OsBADH2*) using TALENs and the CRISPR/Cas9 system can enhance the flavor of rice grains [97,99]. A low-gluten non-transgenic wheat line was developed by knocking out the most conserved domains of the α-gliadin family members, which decreased the genetically predisposed intestinal immune response [100]. Recently, there has been increasing interest in improving the nutritional and health-related traits of crop plants. The CRISPR/Cas9 system has been used to improve the nutritional composition of crops, which has led to increases in the oil content of soybean, the starch quality of potato and gluten-free wheat, the lycopene and γ-aminobutyric acid content in tomato, the carotenoid content of rice, and the yield of high-oleic-acid soybean [101,102,103]. Using an HDR-based method, Dahan et al. edited the *CRTISO* gene in tomato, which significantly increased the carotene content [91]. Cermak et al. inserted the *Cauliflower mosaic virus 35S* promoter into the promoter region of the tomato *ANT1* gene via a homologous recombination involving the twin virus replicator to specifically activate *ANT1* expression and increase the fruit anthocyanin content by several fold. Additionally, tomato fruits with increased post-harvest longevity have been developed using CRISPR/Cas9 technology [104,105]. Japan was the first country to commercialize a genome-edited tomato product with high γ-aminobutyrate contents in September 2021. Jing et al. (2021) used the CRISPR/Cas9 system to knock out *GmFATB1*, which encodes a fatty acyl carrier protein thioesterase that can significantly decrease the abundance of two saturated fatty acids in soybean mutants [106]. The examples of genes from various crops that have been modified by the CRISPR/Cas9 system to increase crop yield and quality were shown in Table 2.

**Table 2 ijms-23-10442-t002:** Examples of genes from various crops that have been modified by the CRISPR/Cas9 system to increase crop yield and quality.

Crop Name	Gene Name	Gene Function	Editing Methods	Mutant Features	References
rice	*PYL1, PYL4, PYL6*	regulated plant growth	Knockout	promote rice growth and productivity	[83]
rice	*GW2, GW5 and TGW6*	negative regulators controlling yield-associated characteristics of rice	Knockout	increased grain weight	[84]
rice	*OsAAP3*	an amino acid osmotic enzyme related to nutrient allocation	Knockout	higher tiller number and grain yield	[85]
rice	*OsSNB*	regulates flower organ development and rice grain shape	Knockout	increased the grain length, grain width and 1000-grain weight	[87]
wheat	*TaGASR7*	grain length and weight	Knockout	1000-grain weight	[88]
wheat	*TaGW2*	encoding RING E3 ligase	Knockout	increased the length and width of wheat grains	[89,90]
wheat	*TaCKX2-D1*	encoding cytokinin oxidase/dehydrogenase	Knockout	Increased grain number and wheat yield	[91]
rice	*OsLOGL5*	Cytokinin activating enzyme	Knockout	increased grain number and weight per spike as well as the yield of rice	[92]
rice	*Wx*	encoding starch synthase	Knockout	reduced the content of amylose content	[25]
maize	*Wx1*	encoding starch synthase	Knockout	Increased maize amylopectin content close to 100%	[49]
rice	*SBEI, SBEIIb*	Determined the fine structure and physical properties of starch	Knockout	increased AC and RS content	[95,96,97]
spring barley	*Waxy*	catalyzed synthesis of amylose	Knockout	reduced amylose content	[98]
rice	*OsBADH2*	encoding betaine aldehyde dehydrogenase	Knockout	increased the flavor of rice	[99]
tomato	*ANT1*	regulated plant growth	in-situ site-specific activation	Increased anthocyanin content	[105]
soybean	*GmFATB1*	encoding FATB protein	Knockout	reduced the contents of two saturated fatty acids in soybean	[106]

### 4.5. Improvement of Crop Abiotic Stress Tolerance

Abiotic stresses, such as drought, salinity, high temperatures, and soil pollution, severely affect crop growth and greatly hinder efforts to increase crop yield and quality. Research conducted to improve the abiotic stress tolerance of crops through CRISPR/Cas9-mediated gene editing is progressing rapidly. A mutation to the maize *ZmSRL5* gene, which is associated with the formation of the maize cuticle wax structure, can enhance maize drought tolerance [107]. Editing the promoter region of *ZmARGOS8*, which encodes a negative regulator of the maize response to ethylene, can also positively affect drought tolerance [108]. Zhou et al. used CRISPR/Cas9 technology to decrease the sensitivity of the elite rice restorer line ‘Hua Zhan’ to abscisic acid and minimize the leaf water loss rate, leading to increased tolerance to drought, high temperatures, and osmotic stress [84]. Kumar et al. mutated the drought tolerance-related gene in *indica* rice using CRISPR/Cas9; the mutants were moderately tolerant to osmotic stress and highly tolerant to salinity stress at the seedling stage, implying their method was appropriate for improving the drought and salinity tolerance of *indica* rice varieties [109]. Lou et al. (2017) modified the *OsSAPK2* sequence and observed that the homozygous T_1_ *OsSAPK2* mutants were relatively insensitive to abscisic acid, but highly sensitive to drought stress, reflecting the relationship between *OsSAPK2* and the drought tolerance of rice. In terms of the cold tolerance of rice, Shen et al. (2017) used CRISPR/Cas9 technology to specifically edit *OsAnn3*; the subsequent phenotypic analysis indicated the six mutant lines were more sensitive to low-temperature stress than the wild-type control [110]. The pollution of arable land during industrialization and urbanization is a serious concern. Preventing the accumulation of toxic heavy metals in food crops is critical. Accordingly, rice lines with decreased radioactive cesium, arsenic, and cadmium contents have been generated by deleting *OsHAK1*, *OsARM1*, and *OsNramp5*, respectively, using CRISPR/Cas9 technology [111,112,113].

### 4.6. Improvement of Other Crop Traits

In addition to stress resistance/tolerance as well as traits related to yield and quality, CRISPR/Cas9 technology has also been used to improve other crop traits (e.g., fertility), leading to the development of novel plant types and haploid materials.

The creation of male-sterile materials is extremely important for hybrid seed production. Researchers have conducted a series of investigations regarding the editing of pollen fertility genes using CRISPR/Cas9 technology. Li et al. used the *T. aestivum TaU3* RNA polymerase III U3 promoter to drive the optimized CRISPR/Cas9 vector. Three homologous alleles encoding the wheat redox enzyme NO POLLEN 1 (NP1) were edited to produce fully male-sterile wheat mutants [114]. Chen et al. constructed a CRISPR/Cas9 vector to delete *Male sterility gene 8* (*MS8*) in maize. The resulting mutant exhibited a male-sterile phenotype, which was consistent with Mendelian genetic laws and was stably inherited by the later generations [115]. Rice two-line male-sterile lines have been divided into photosensitive male-sterile lines and thermosensitive male-sterile lines. Li et al. edited the carbon starvation gene *CSA* in the pollen grains of rice variety ‘Kongyu 131’ and reported that the *csa* mutant exhibited a male-sterile phenotype under short-day conditions and a male-fertile phenotype under long-day conditions (i.e., photosensitive nuclear male-sterile mutant) [116]. Huang et al. targeted the *TMS5* gene in rice. The *TMS5* mutant was completely male-sterile at high temperatures and male-fertile at low temperatures; the transition temperature for the pollen fertility of the *TMS5* mutant was 28 °C [117]. Shen et al. generated photosensitive/thermosensitive male-sterile lines by using CRISPR/Cas9 to modify the *Photoperiod-thermosensitive genic male-sterile 2-2* (*PTGMS2-2*) gene [118].

The CRISPR/Cas9 system has been used to edit *ZmMTL* (*ZmPLA1*) to generate maternal haploid inducers with strong haploid identification markers applicable for the breeding of doubled-haploid cereals, including maize [119].

Rice plant type is an important factor affecting grain yield and quality. The primary traits that determine the plant type are plant height, leaf type, tiller number, tiller angle, and panicle type. Hu et al. used the CRISPR/Cas9 system to edit the semi-dwarf gene *SD1*, which decreased the plant height of the *sd1* mutant by about 25% [120]. Li et al. used CRISPR/Cas9 technology to edit the upright panicle-type gene *DEP1* and the ideal plant-type gene *IPA1* in the ‘Zhonghua 11’ rice variety. The *dep1* mutant was characterized by an upright plant and compact panicle phenotype. The number of tillers either increased or decreased in the *ipa1* mutant, reflecting the two extreme phenotypes induced by the mutation [121].

## 5. Technical Problems Associated with the CRISPR/Cas9 System and Potential Solutions

The CRISPR/Cas9 system and its associated gene-editing techniques have developed rapidly in recent years, with many reports describing the functional annotation of crop genes and the highly precise molecular breeding of crops. However, there are still certain limitations to the available CRISPR/Cas9 editing technology, including the occurrence of off-target effects, editing scope, and limitations associated with the plant genetic transformation system.

### 5.1. Off-Target Effects

The cleavage of non-target genomic sites by Cas9 leads to off-target effects. Two factors influence the occurrence of such effects. First, sgRNA may bind to non-target sequences. The specificity of the binding of sgRNA to its target sequence is critical for the success of CRISPR/Cas9 methods. However, because of the complexity of genomes, in addition to the target site, sgRNA may also bind to similar sequences. Thus, the localized activation of the Cas9 endonuclease will lead to the cleavage of non-target sites, resulting in off-target effects. Second, Cas9 may detect a non-standard PAM. In the CRISPR/Cas9 system, Cas9 should cut three bases upstream of the PAM site. However, sometimes Cas9 recognizes the standard PAM near the target site as well as a non-standard PAM, resulting in off-target effects. Therefore, non-standard PAMs must be considered when designing the target sequence to decrease the possibility of off-target events [122].

The Cas9-related off-target effects may be limited by increasing the specificity and fidelity of Cas9. Studies on the off-target effects of CRISPR/Cas9 involving GUIDE-Seq and Digenome-Seq have shown that the genome-wide DSBs caused by Cas9 can be analyzed to predict off-target sites [123,124]. However, identifying DSBs in the genome caused by the CRISPR/Cas9 system remains challenging.

Some researchers have been able to restrict the off-target effects of SpCas9 by mutating certain enzyme regions [125]. Specifically, Asn 497, Arg 661, Gln 695, and Gln 926 in SpCas9 were converted to Ala to obtain SpCas9-HF, after which GUIDE-Seq was used to analyze off-target sites throughout the genome. The results indicated that there were significantly fewer off-target events for SpCas9-HF than for SpCas9 [126]. The Cas9 protein contains multiple domains with different functions. We mutated the SpCas9 REC3 domain, which recognizes the complementary strands formed by sgRNA and target sequences, controls the HNH nuclease, and regulates the overall catalytic activity, to generate HypaCas9 with increased specificity [127]. Optimizing the Cas9 structure can also enhance its specificity [125]. Moreover, shortening the in vivo Cas9 activation time can limit the off-target effects. For example, the RNP complex of sgRNA and Cas9 can be delivered directly into the cell or Cas9 can be split into two parts and then induced by small molecules to recombine into the intact Cas9 protein within the cell [128,129].

### 5.2. Target Site Limitations

The CRISPR/Cas9 system has accelerated the improvement of crop traits, but the target sites of the CRISPR/Cas9 nuclease are limited by the corresponding PAM sequences, making it difficult to edit all target loci [90]. Therefore, appropriately modifying Cas9, increasing the compatibility between the CRISPR nuclease and different PAMs, and expanding the genomic region editable by CRISPR/Cas9 will influence whether the CRISPR system can be widely adopted to alter crops in the future. Researchers should aim to increase the target range of Cas9 by extending the PAM recognition range. In 2015, Keith’s laboratory developed a mutant SpCas9-VRQR that recognizes the NGA sequence as well as a mutant SpCas9-VRER that recognizes NGCG [130,131]. Variants that recognize different PAM sequences have subsequently emerged, leading to a significant increase in the genomic range that can be edited by Cas9. The XCas9 3.7 variant developed by David Liu’s laboratory in 2018 effectively recognizes NGG, GAA, and GAT, thereby greatly increasing the target range of Cas9 [132]. Nureki’s laboratory developed a variant of SpCas9 that recognizes NG, further expanding the target range [133]. Subsequent research has further expanded the PAM recognition region beyond the existing range. In 2020, David Liu’s laboratory constructed a series of SpCas9 mutants that added NRNH to the recognizable PAM sequences [134]. The modification of SpCas9 by the Kleinstiver laboratory generated the mutant SpRY, which can recognize NRN and NYN [135]. Therefore, innovations in the technology have essentially made all PAMs recognizable by SpCas9 and its mutants, which has broadened the scope of crop gene editing. However, this expansion has been linked to a decrease in editing efficiency. Consequently, the editing efficiency will need to be further optimized. Several studies have demonstrated that expanding the editing scope leads to self-targeting, which contributes to the decrease in editing efficiency and the limited applicability of Cas variants. Resolving this problem will further increase the utility of the CRISPR system [80]. Additionally, FnCas9 has been optimized in terms of its protein structure to recognize a different PAM sequence [136].

### 5.3. Foreign Genes

The *Agrobacterium*-mediated transformation of plants with CRISPR/Cas9 gene-editing vectors may result in the random integration of vector fragments into plant genomes, leading to the introduction of foreign genes and potential biosafety issues [137]. For crops that reproduce sexually, lines lacking exogenous genetic material can be screened via the genetic separation of sexual generations. However, for crops that reproduce asexually, the isolation and removal of foreign genes via genetic separation is impossible [138].

If the RNP complex is used to avoid importing foreign genes, it can be applied to crops without the need for the genetic separation of sexual generations. However, this method is relatively difficult to complete, which necessitates further technical optimizations to increase its efficiency and utility for additional crops. Transgene-free plants can be obtained by RNP transfection, the transient expression of transgenes, and nanobiotechnology-based approaches [139]. Furthermore, an effective tool for assessing the biological safety of genome-edited products has been developed. An online tool for detecting foreign elements in genome-edited organisms is available. It can be used when there is no information regarding the foreign carrier component. This tool screens whole-genome sequencing data to simultaneously detect 46,695 exogenous components [140].

### 5.4. Limitations in Genetic Transformation Systems

The editing of crop genes using the CRISPR/Cas9 system is dependent on an efficient and stable genetic transformation system. *Agrobacterium*-mediated transformation and gene gun bombardment remain the main methods used for crop genetic improvement. *Agrobacterium tumefaciens* is a gram-negative soil bacterium. In its natural environment, *A. tumefaciens* can infect wounded dicotyledonous plants, after which its T-DNA is integrated into the host genome and expressed through the host DNA repair mechanism [141]. *Agrobacterium*-mediated transformation exploits this DNA transfer mechanism to introduce foreign DNA into plant genomes. Given its dependence on a tissue culture regeneration system, the *Agrobacterium*-mediated method is not suitable for all crops and tissues. Gene gun bombardment is a DNA transformation method that uses high-pressure gas to incorporate foreign genes coated on the surface of metal particles into recipient cells. The gene gun bombardment method facilitates the simultaneous insertion of multiple genes, RNA sequences, or proteins to modify plant genomes. However, owing to the limitations in the established tissue culture regeneration systems, the utility of gene gun bombardment methods is relatively limited and the transformation efficiency is low. Although other methods have been used to transform crops, including methods involving polyethylene glycol, liposomes, silicon carbide, and microinjections, they cannot be broadly used because of their limitations (e.g., tissue culture regeneration system and genotype dependence) and they are time-consuming and expensive [142].

A new magnetic nanobead-based transformation method may be a viable option. The method, which was developed by Wang et al. to transform maize using pollen, is not dependent on the genotype and transformation regeneration system. Additionally, it is highly efficient and relatively simple, in part because it only requires magnetic nanobeads and a magnetic plate. Hence, the transformation of pollen grains to modify the maize genome may be completed within several hours and can be performed in the field. This novel transformation system may be applicable for crops lacking an established tissue culture regeneration system (e.g., some fruit trees and other horticultural crops) [143]. This method may expand the range of crops that can be genetically modified by the CRISPR/Cas9 system. Future studies should aim to develop more convenient and efficient methods for transforming plants.

## 6. Summary and Outlook

Crop improvement through domestication is a very slow process. Nevertheless, approximately 7000 crops were cultivated worldwide during the domestication stage, which laid the foundation for the cultivation of modern varieties Cross breeding was initiated in the mid-to-late 19th century. Breeders and scientists purposefully selected different parents for hybridizations (e.g., crossing, self-crossing, and backcrossing) to generate crop varieties that combine the desired traits of both parents. Breeding on the basis of heterosis and active mutagenesis emerged toward the end of the 19th century. The techniques for cross breeding, heterosis breeding, and active mutagenesis breeding were instrumental for traditional breeding [144]. These breeding methods have greatly increased crop yields and alleviated the food shortage caused by the substantial increase in the global population over the past century, but they still rely on breeders selecting materials according to phenotypic examinations. Moreover, the utility of conventional breeding techniques may be somewhat limited for complex traits. Therefore, breeding new varieties with high yields and quality as well as stress resistance is a major challenge. To complement traditional breeding techniques, maintain sustainable agricultural production, and create new crops that efficiently use environmental resources (e.g., nutrients and water) and tolerate biotic and abiotic stresses, molecular breeders have developed and applied gene-editing technologies. Therefore, the emergence of gene-editing technology should supplement rather than replace traditional breeding methods.

The cost and efficiency of gene-editing technology mainly depend on two factors. First, specific techniques (e.g., transformation systems) must be developed and optimized. The establishment of low-cost, low-risk, and efficient transformation systems according to different crop characteristics will increase the applicability of gene-editing technology. The second factor is government regulatory policies. More specifically, there is an international debate over whether CRISPR-edited varieties should be regulated in the same way as traditional genetically modified crops or be allowed to enter the market without any regulations. For example, the USA and the European Union evaluate CRISPR-edited crops using very different regulatory frameworks, but most countries still apply the existing policies regulating genetically modified crops. The international community has been considering two important questions related to CRISPR-edited crops. First, is it possible to exclude certain CRISPR-edited crops from regulatory oversight? Second, what safety-related data would be needed if CRISPR-edited crops are to be regulated in a given country? The amount of safety-related data required will affect the overall cost of the regulation, which is an important factor to consider when commercializing new CRISPR-modified plants [145].

In addition to the effects of climate change and population growth, the sustainable production of food faces many obstacles as urbanization decreases the available arable land area [1]. Therefore, revolutionary plant breeding techniques, including CRISPR/Cas9, are needed to generate new traits and varieties that can increase yields or tolerate adverse conditions. Researchers have successfully modified and improved many quality-related traits using the CRISPR/Cas9 system. Moreover, some gene-edited crops have been commercialized, including TALEN-*fad2* soybean, TALEN-*ppo* potato, and CRISPR-*wx1* maize, suggesting that the associated technology has advanced beyond the proof-of-concept research stage [94].

In this review, we discuss the utility of the CRISPR/Cas9 system and its derivatives for crop genetic improvement. However, owing to the limited space available, many studies that have contributed to this research field are not mentioned in detail. With the continuing development of sequencing technology and the decrease in sequencing costs, genome sequencing data for an increasing number of crops are now available, which provides the foundation for editing the genes of additional crops in the future. CRISPR/Cas9 technology, which enables the rapid and precise editing of genes in diverse crops, has contributed to increases in crop yield, quality, disease resistance, and many other phenotypic characteristics. Furthermore, its application has led to decreases in the use of fertilizers and pesticides, human labor, and the consumption of water. Thus, this technology will promote sustainable agricultural development and play a vital role in solving future food insufficiency-related crises.

## Figures and Tables

**Figure 1 ijms-23-10442-f001:**
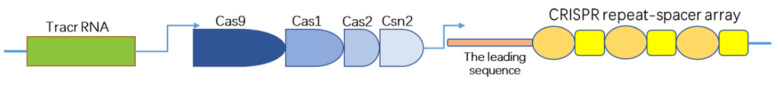
Basic CRISPR/Cas9 structure. The CRISPR/Cas9 locus consists of three elements, namely a tracrRNA at the 5′ end, a Cas protein-encoding gene, and a CRISPR locus at the 3′ end, which includes a lead sequence, a 23–50 bp repeat sequence, and a spacer sequence.

**Figure 2 ijms-23-10442-f002:**
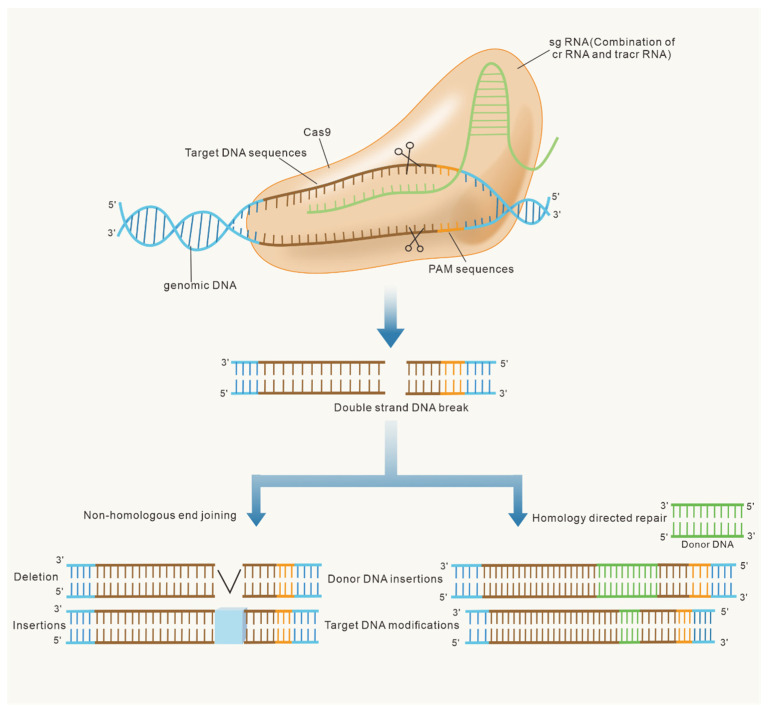
Genome editing at the CRISPR/Cas9 target locus. Site-specific nucleases introduce double-strand breaks where genes are modified by two repair pathways. Non-homologous end joining (NHEJ) knocks out genes via deletions or insertions in the absence of donor DNA. Homology-directed repair (HDR) results in the insertion of donor DNA on the basis of homologous regions and the correction of gene sequences according to small changes in either DNA strand.

**Table 1 ijms-23-10442-t001:** Examples of genes from various crops that have been modified by the CRISPR/Cas9 system to increase disease resistance.

Crop Name	Gene Name	Gene Function	Editing Methods	Mutant Features	References
barley	*MLO*	reduced resistance to powdery mildew	knockout	improved resistance to powdery mildew	[51]
wheat	*TaMLO-A1*	reduced resistance to powdery mildew	knockout	improved resistance to powdery mildew	[52]
tomato	*SlMLO1*	reduced resistance to powdery mildew	knockout	improved resistance to powdery mildew	[53]
tomato	*MLO1*	reduced resistance to powdery mildew	knockout	improved resistance to powdery mildew	[54]
tomato	*EDR1*	encoded MAPKKK protein kinase	knockout	improved resistance to powdery mildew	[54]
rice	*OsERF922*	involved in the modulation of multiple stress tolerance	knockout	enhancing blast resistance	[56]
rice	*OsSEC3A*	interacted with rice SNAP25-type SNARE protein OsSNAP32 and phosphatidylinositol-3-phosphate	knockout	enhanced resistance to the fungal pathogen Magnaporthe oryzae	[55]
grape	*VvWRKY52*	play roles in biotic stress responses	knockout	increased the resistance to Botrytis cinerea	[57]
rice	*SWEET11, SWEET1113, SWEET1114*	transporter genes required for disease susce	knockout	increased broad spectrum resistance to different physiological races of Xoo	[58]
grape	*CsLOB1*	a critical citrus disease susceptibility gene	editing the PthA4 effector binding element	increased canker-resistance	[59]
grape	*CsLOB1*	a critical citrus disease susceptibility gene	remove the effector binding region of CsLOB1	enhanced resistance to Xcc	[60]
apple	*DIPM-1, DIPM-2,* *DIPM-4*	disease susceptibility genes	knockout	increased resistance to fire blight disease	[61]
potatoes	*StDND1, StCHL1, StDMR6-1*	disease susceptibility genes	knockout	increased resistance against late blight	[67]
chili pepper	*CaERF28*	disease susceptibility genes	knockout	increased anthracnose resistance	[68]

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
