# Peer review of "CRISPR/Cas9 Technology and Its Utility for Crop Improvement"

_ijms, 2022, doi:10.3390/ijms231810442_

Round 1

Reviewer 1 Report

Dear Authors,

The manuscript "Recent advances in CRISPR/Cas9 technology for accelerated crop improvement" clearly presents aspects of the use of CRISPR/Cas9 technology in plant breeding. It is an interesting addition to the knowledge for those involved in this research topic. It will also be an interesting reading for those who are new to this topic.  In Section 5.4, I suggest referring readers to the review paper :https://doi.org/10.3390/ijms22084206.

I have the impression that the font in the tables does not follow the journal template.

Reviewer 2 Report

I carefully read the review manuscript. This manuscript attempts to summarize the recent advances in CRISPR/Cas technologies and their use for crop improvement. Several aspects in the current draft are incorrectly portrayed, a description of several recent CRISPR-based tools is missing, and the content needs extensive revision for language and accuracy. Also, I hardly find any insightful discussion about this topic in it. Here are some concerns:

Major-

1.      Section 2 and 3- Considering the title and scope of the review, lengthy paragraphs about CRISPR history, Figure 1, and mode of action in bacteria seem unnecessary. The authors should revise the text in sections 2 and 3, particularly considering the plant application perspective.

2.      The review title constitutes the- advances in CRISPR/Cas9 technology for crop improvement. I hardly find the description of advances in CRISPR technologies, and a section about this aspect would add more value to the paper.

3.      Line 95-97 - We designed a fusion guide RNA (fgRNA) that fused both the 5’-scaffold sequence of Cpf1.

-          The fgRNA details seem irrelevant to the other points discussed in this section.

4.      The current representation of the CRISPR/Cas9 system in Figure 2 is short of meeting the criteria of scientific illustration. The authors should improve the overall quality of the Figure.

5.      Also, in Figure 2, the model of the CRISPR/Cas9 system is wrong; the targeted strand should be in the direction of 3'-5'.

6.      Line 147-148 - What is the relevance of the animal aspect? Authors may refer to recent papers describing the CRISPR use in plants (like Pramanik et al., 2021 https://doi.org/10.1016/j.molp.2020.11.002) rather than describing animal related CRISPR uses.

7.      Line 155 - XCas9 or xCas9? Also, nickase Cas9 (nCas9) would be a better choice in this sentence. There are two versions of nCas9 – (D10A) and (H840A). Discussion about this aspect which is crucial for BE and PE tool use at the appropriate place, would be ideal.

8.      Section- Application of CRISPR/Cas9 in gene editing-

What is the logic of dividing this section into subheadings? Tool or outcome after using a tool? The authors should clearly state it at the beginning of this section. If the basis of categories is about a result (application), why are tools mainly described in the subsections? The description and flow of content are confusing, and it would be ideal to describe tools and applications separately.

9.      Line 169-170 - One or more sgRNA target sites can be designed for each target gene.

-          The meaning is not clear.

10.  Section 4.2. The description of gene knock-in is not clear to me. One of the purposes of knock-in is to facilitate the allele replacement or targeted transgene insertion. Authors may reconsider the context in the section.

11.  Line 180-181 - Glutathione, for example, is a region of the chromosome…?

-What do authors mean to say…?

12.  Section 4.3. Two CBE tools were published in 2016 and are used frequently in animals and plants (based on APOBEC and PmCDA1). PmCDA1-based tool and its citation should be described in this section. Also, crucial BE tools are missing, for example, CGBE and ddCBE.

13.  Line 210 - The CBEs and ABEs can convert C to T or A to G bases but cannot convert any base?

- What do authors mean to say…?

14.  Application of CRISPR/Cas9 in crop breeding – Rather than a collection of some examples, the authors should update this section with recent papers and provide insightful discussion about how CRISPR use is advancing the six research areas that the authors choose to divide this section.

Minor-

I show some examples of the minor aspects which should be examined in this manuscript.

1.      Line 42, 82, 156 – add space between text and reference. This aspect should be revised throughout the manuscript.   

2.      Line 127 - PAM? Spell out the full form while using the term the first time.

3.      Line 128 - The PAM is usually composed of three bases (NGG; N is any base). This statement is partially incorrect. SpCas9 recognizes NGG, and other Cas9 orthologs (NmeCas9, CjCas9, and others) have variable PAMs. Consider rephrasing it.

4.      Line 136 - fracture site? The authors should consider using relevant scientific words from molecular biology.

5.      Line 160- CRISPR-Cas9 or CRISPR/Cas9?

6.      Line 163- guided editing?

Reviewer 3 Report

This review describes CRISPR technology for enhancing crop production. Authors have tried to include its various applications to improve crop yield. My comments

- Provide various approaches of traditional breeding have been used and have necessitated the adoption of new technologies.

- Can you get rid of crop breeding by adopting this technology?

- This technology requires gene transfer with Agrobacterium/blastic method and may raise ethical issues

- The role of plant tissue culture for plant regeneration is crucial for success and may have limitations to use widely in different crops.

- Cost-effectiveness of this technology in crop production?

Is this technology an additional tool for breeders just like genetic transfer methods? How long it will take to become a regular tool to breed new crops? From the breeder's point of view, its usage is practical or just an academic exercise?

Round 2

Reviewer 3 Report

Minor changes in English is needed

Author Response

The english text was edited by Liwen Bianji Compony. 
